# Differentiating Breast Myopathies through Color and Texture Analyses in Broiler

**DOI:** 10.3390/foods9060824

**Published:** 2020-06-23

**Authors:** María del Mar Campo, Leticia Mur, Ana Guerrero, Marta Barahona, Virginia Celia Resconi, Danielle Rodrigues Magalhaes, Eduardo Lisbinski, Bruna Boito, Ivanna Moraes de Oliveira, José Luis Olleta

**Affiliations:** 1Department Animal Husbandry and Food Science, Instituto Agroalimentario IA2, Universidad de Zaragoza-CITA, Miguel Servet 177, 50013 Zaragoza, Spain; leticiamurpalus@hotmail.com (L.M.); aguerre@unizar.es (A.G.); martabm@unizar.es (M.B.); resconi@unizar.es (V.C.R.); d.magalhaes@yahoo.com.br (D.R.M.); lisbinski.utfpr@gmail.com (E.L.); bruna.boito@gmail.com (B.B.); imoraesdeoliveira@yahoo.com.br (I.M.d.O.); olleta@unizar.es (J.L.O.); 2GRUPO UVESA, Pol. Ind. Montes del Cierzo A68 km 86Apdo.-67, 31500 Tudela, Spain

**Keywords:** wooden breast, white striping, spaghetti meat, reflectance, compression test, Warner–Bratzler, texture profile analysis

## Abstract

Wooden breast (WB), white striping (WS) and spaghetti meat (SM) are breast myopathies of the *Pectoralis major* that greatly affect meat quality in broilers. To differentiate color and texture characteristics with instrumental methods, some of them applied for the first time in this species, 300 carcasses were randomly chosen from an abattoir from five different flocks from the same farm, at a rate of 60 carcasses from each flock. Twenty-four hours after slaughter, both side breasts were dissected, and yields calculated. Color was measured on the surface of the breast with a spectrocolorimeter and reflectance values obtained. Texture was measured on raw meat with a modified compression test that hinders the fiber from expanding transversally and a texture profile analysis (TPA) and also on cooked meat with a Warner–Bratzler shear and a TPA. Color differs between severity degrees, increasing redness (from −1.77 to −1.32 in WB) and, especially, yellowness (from 5.00 to 6.73 in WS) and chroma (from 5.75 to 7.22 in SM) with the severity of the myopathy. The subtraction R630 minus R580 was found to be a useful index to differentiate breast myopathies degrees. The modified compression test can be considered an effective tool to assess the hardness of different structures in each myopathy. Texture differences in the myopathies are better assessed in raw than in cooked meat.

## 1. Introduction

Breast myopathy is an important quality problem that causes severe economical losses in the poultry industry [1,2]. The development of modern broilers has facilitated the emergency of three different myopathies in recent years affecting the *Pectoralis major* muscle: wooden breast (WB), white striping (WS) and spaghetti meat (SM). The origin of these myopathies is being studied since they occur more often as the weight of the animal increases. Muscle growth after hatching is mainly due to myofiber hypertrophy increasing length and width [3]. This requires the incorporation of nuclei into the cell from stem cells known as satellite cells [4]. These satellite cells also participate in repairing tissue after damage [5]. Modern broilers are characterized by a fast and very efficient growth, with a fast rate of muscle regeneration and large fiber size suggesting higher satellite cells number and faster growth kinetics at the core of the muscle than those found in low weight selection lines [6], where the occurrence of these myopathies is scarce. White striping occurs when the regeneration is mild and damaged muscle is replaced with adipocytes and fibrosis [7] whereas wooden breast shows accumulation of interstitial connective tissue or fibrosis [8]. Spaghetti meat is characterized externally by a low cohesion of the fiber bundles with a loose connective tissue deposition [9], probably due to the immaturity of the recent incorporated collagen mainly affecting the superficial portion of the muscle [10]. Different husbandry strategies (nutritional composition, feeding regime) have being used without much success to replicate, control and reduce the incidence of these myopathies [11,12].

Besides texture, color is a very important meat trait that drives consumer choice, especially for raw meat, as consumers want to see the product [2]. Nevertheless, less attention has been given to this quality trait in relation with texture when myopathies occur focusing more in appearance and firmness aspects [13] more related to texture. Different texture devices and tests have been used assessing major pectoralis myopathies, such as Meullenet–Owens razor shear [14,15], Allo–Kramer shear [16], Warner–Bratzler shear [17] or texture profile analysis (TPA) which is more commonly used in this species [14,17,18]. We hypothesize the usefulness of a compression test that is being used successfully in other species [19,20], in order to determine the implication of different muscle components in texture differences when a myopathy occurs. The second hypothesis relates to the modification of the color of the breast due to the development of myopathies, which we aim to assess through spectrocolorimeter analyses.

## 2. Materials and Methods

### 2.1. Animal Management and Sampling

This study was performed with animals reared in a farm under commercial practices approved for 39 kg/m^2^ (RD 692/2010), following national regulations in animal welfare (Council Directive 2007/43/EC), with animals being slaughtered between 35 and 41-day-old at a commercial abattoir (UVESA, Tudela (Navarra, Spain)) with electrical stunning in agreement with regulation 1099/2009 of the European Union on the protection of animals at the time of killing.

Through four different flocks with ROSS 308 and one with COBB 500 broilers along 2017, 300 carcasses randomly chosen at the abattoir were assessed, 60 from each flock. Feed withdrawal lasted between 6 and 10 h prior to slaughter including fasting in house, catching, transport and lairage times, where the transport from the farm to the slaughterhouse was less than 30 min. Carcasses were defeathered, eviscerated, headless and vent after feed removal prior to chilling. At 24 h after slaughtering, carcasses were transported under commercial refrigerated conditions at 4 °C from the processing plant of the abattoir to the facilities of the meat quality lab at the veterinary faculty of the University of Zaragoza. They were then weighed and both side breasts were dissected with a knife in order to calculate the breast yield in relation to the cold carcass. Afterward, the degree of WB, WS and SM myopathies was visually assessed on a 3-point scale: 0, absence; 1, moderate; 2, severe [21]. None of the breasts was found severe for SM.

### 2.2. pH

At 24 h postmortem, pH was measured with a penetrating electrode Labprocess PC7 XS on the caudal portion of the left breast.

### 2.3. Color

Color was measured on the surface of the cranial portion of the left breast with a reflectance spectrocolorimeter Minolta CM 2002 in the CIEL*a*b* space [22], with an illuminant D65 and a 10° standard observer. The average of three measurements in three different points of the surface of the *Pectoralis major* muscle, avoiding dirty areas with evident hemorrhages or liquid, was obtained. Lightness (L*), redness (a*) and yellowness (b*) were recorded and chroma (C*, $\sqrt{a^{2}+b^{2}}$) and Hue angle (H°, tan^−1^ (b*/a*) in degrees) were calculated. Reflectances between 360 nm and 740 nm were recorded at 10-nm intervals, selecting reflectance at 630 and 580 nm to calculate reflectance indexes: R630-R580 and R630/R580. Then, each breast side was individually vacuum packaged and immediately frozen at −18 °C until texture analyses were performed within a month after slaughter.

### 2.4. Texture in Raw Meat

The left side breast was used for raw meat analyses. Meat was thawed without breaking the packaging at 4 °C for 24 h. Starting from the cranial part of the *Pectoralis major* muscle, 1-cm-thick slices were cut, and then, in these slices, 1-cm^2^ section parallelepipeds were obtained. They were analyzed through a compression test with 1-cm^2^ contact surface, in which the muscle fibers can only expand longitudinally [19,23] with an INSTRON 4301. Compression at 20% and 80% of the total compression rate, as well as the total compression rate, expressed in N/cm^2^, were recorded. At the same time, some of the parallelepipeds were used to obtain 1-cm^3^ samples to be assessed with a texture profile analysis (TPA) in a TA-XT2i, with the direction of the compression force perpendicular to the direction of the fibers, as happened in the previous texture analysis. Hardness, adhesiveness, cohesiveness, springiness, gumminess, chewiness and resilience were calculated.

### 2.5. Texture in Cooked Meat

The right-side breast was used for cooked meat. After thawing the meat in the same conditions as previously mentioned, cooking was performed in a water bath at 75 °C until reaching an internal temperature of 70 °C measured with an internal thermocouple, without vacuum, but without direct contact between the meat and the water. Slices and parallelepipeds were obtained in the same way as in raw meat texture analysis after cooling at room temperature. A Warner–Bratzler device was applied, also with an INSTRON 4301, and shear force was obtained. In addition, 1 cm^3^ samples were obtained and subjected to a TPA in a TA-XT2i with the same methodology and variables as previously explained for raw meat.

### 2.6. Statistical Analysis

A general lineal model was used with SPSS 26.0 to assess differences in carcass, pH, color and texture variables in the different tests, with each myopathy as a fixed effect and flock as a block effect. Tukey’s multiple range test was used to find differences between mean values. Individual carcasses were considered the experimental unit. A discriminant analysis was also applied using the selecting λ_Wilks_ parameter with stepwise to determine the most significant wavelengths’ reflectances in detecting different severity of the different myopathies, in order to classify the samples at each myopathy level.

## 3. Results

### 3.1. Carcass and pH

From the 300 carcasses assessed in this trial, 55.6% of the carcasses showed signs of WB, being 11.3% of carcasses with severe WB. The occurrence of WS affected 41.7% of the carcasses, having 11.7% of the carcasses severe WS. In the case of SM, none of the carcasses showed severe level of this myopathy, having 5.3% of the cases moderate SM.

Carcass weight increased from 1.92 kg to 2.06 kg with the appearance of both Wooden breast (*p* < 0.001) and White striping (*p* = 0.010). Nevertheless, the presence of Spaghetti meat was not associated with differences in carcass weight (Table 1). The presence and severity of WB and WS myopathies was especially related to higher breast yields (*p* < 0.001). The absence of myopathy had an average breast yield of 31.3% in both myopathies, differing significantly from moderate degree (32.1% and 32.4% for WB and WS, respectively) and again from severe degree (33.2% and 33.6%, respectively), implying that an average increase of 1% yield would show an increment of one level of each myopathy (*r* = 0.28 and *r* = 0.35, respectively, *p* < 0.001). This increase did not apply to the presence of SM.

No significant differences were found in pH due to different levels of Spaghetti meat (Table 1). However, a severe degree of WB or WS was found to be associated with higher pH values (*p* = 0.007; *p* = 0.018).

### 3.2. Color

Significant color differences have appeared with different levels of each myopathy, especially WB and WS (Table 2). Severe WB showed lighter meat (*p* = 0.005). The presence of WB increased redness from −1.77 up to −1.32 (*p* = 0.001), yellowness (*p* = 0.005) and chroma* (*p* = 0.042), reducing Hue angle (*p* = 0.014). The presence of WS also increased redness (*p* = 0.005) from −1.74 up to −1.17, but each increment in the degree of this myopathy showed higher lightness (*p* = 0.005), yellowness (*p* < 0.001) and chroma (*p* < 0.001) and less H° (*p* = 0.002). The presence of moderate SM only showed differences in b* value and chroma*, with higher yellowness (*p* = 0.002) and chroma* (*p* = 0.003) when the myopathy appeared at this level.

The index R630-R580 showed important differences in all myopathies, especially WB and WS (*p* < 0.001). In this way, the absence of WB, WS and SM were clearly separated from moderate or severe myopathies. The index R630/R580 showed differences only in WB (*p* = 0.009), in the same line as R630–R580.

Only 64.7% of samples without WB were correctly classified as such (Table 3) when using the whole spectra. The correct classification of samples according to the 6 wavelengths that were finally included in the discriminant analysis after 12 steps was reduced to 53.4% of the samples with moderate WB and only 2.9% of breasts with severe WB. In the case of WS, 92.0% of samples without this myopathy were correctly classified after 3 steps with only 3 reflectances: 360, 610 and 700. In the case of SM, all samples without this myopathy were classified as such with just 2 reflectances: 420 and 630 after 2 steps in the analysis, which represents 94.7% accuracy of the total samples.

### 3.3. Texture in Raw Meat

Significant differences were found in raw meat with the compression test at different levels of WB and SM, but not in WS (Table 4). Compression at a rate of 20% of total compression showed an increase from 11.4 N/cm^2^ to 13.1 N/cm^2^ with the severe degree of WB (*p* < 0.001). Although no differences were found among the degrees of WB at 80% of the total compression rate, at the full rate of compression a severe degree of WB had higher values (24.7 N/cm^2^) than meat with absence of this myopathy (22.1 N/cm^2^, *p* = 0.016). SM showed a different tendency, since the presence of this myopathy was associated with lower values for a compression rate of 80% of the total compression (15.6 vs. 12.1 N/cm^2^, *p* = 0.029) or for the full rate of compression (23.0 vs. 18.7 N/cm^2^, *p* = 0.032).

The texture profile analysis has also shown higher hardness in meat with a severe degree of WB (*p* = 0.016) together with lower cohesiveness (*p* = 0.001). This characteristic also applies to the degrees of WS, since the severe WS was the least cohesive (*p* = 0.007). The rest of characteristics of the TPA have not been modified according to the level of myopathy, except for a tendency of SM to show a higher springiness (*p* = 0.081).

### 3.4. Texture in Cooked Meat

Differences in texture due to the different degrees of myopathy were less evident in cooked meat than in raw meat. Warner–Bratzler is a widely used cell to assess texture, but only a tendency (*p* = 0.087) of the most severe degree of WS to have less shear force was found (Table 5). No significant differences have arisen between different levels of WB or SM myopathies.

With regards to TPA, only cohesiveness was significantly lower with severe WB, as previously mentioned in raw meat. But those differences previously found in hardness or in WS in raw meat have disappeared after cooking. On the contrary, springiness showed a tendency (*p* = 0.062) to decrease when SM appears.

## 4. Discussion

A new approach to the study of breast myopathies through differences in color reflectance and a compression texture device that had not been previously used in poultry was done.

### 4.1. Carcass and pH

The appearance of breast myopathies is a cause of important economic losses to the industry [2] deriving meat for further manufactured products with lower value than fresh meat. The occurrence of WS (41.7%) in the current study coincides with the 43% found by Lorenzi et al. [24] in 70 flocks at the slaughterhouse, but it is lower than the 55.8% found by Kuttapan et al. [25] although in older animals (59 to 63-day-old) or the 70% found in females of 46-day-old [26]. The incidence in heavy weight flocks can be higher, reaching 57.4% [24] or even 82.5% in 55-day-old males [26]. Heavy flocks are slaughtered at a live weight between 3.0 and 4.2 kg. But common practices in most of Spain target 2.7 kg as the slaughter weight of the animal, considering these flocks as medium weight. This target is achieved after 38–42 days of rearing and this decreases the emergence of myopathies. Differences in live weight have already being considered as a cause for differences in the occurrence of myopathies that are not consistent among studies [12,25].

Increasing breast yield increases the possibility of developing breast myopathies [24]. This aspect is strain dependent with a higher incidence of WS in high breast yield hybrids [24]. We have observed that each increase around 1 percentage point from 31.3% of breast yield, increases by one degree the severity of WB and WS.

Severe WS breasts have showed higher pH as previously found by other authors [11]. A reduced carbohydrate metabolism was proposed for the lower glycolytic potential and higher ultimate pH [27].

### 4.2. Color

Color is an important criterion for consumers at purchase. The instrumental measurement of color can be an important tool for the industry as a non-destructive method. The higher lightness as the severity of the myopathy evolves may be related to changes in the muscle structure. Furthermore, pH increases with the myopathy as seen before. Higher pH s were related to darker meat in other species, therefore lower L* values due to lower reflected light and higher absorption at all wavelengths [28] would be expected. At high pH, the muscle is able to retain more intracellular water absorbing more wavelengths and reflecting less light. At low pH, proteins reach their isoelectric point, when water would not be retained. However, when breast myopathy occurs, an inflammatory process appears and there is higher moisture in the muscle [16,18,29] due to the presence of edema as a consequence of the inflammatory process [18]. At the same time, less protein is present in the breast [16,18,29] which implies less possibility to bind water at intracellular level. In fact, distribution and mobility of water fractions are altered, with higher extra- and lower intra-myofibrillar fraction [29]. In addition, higher drip loss and cooking losses have been described as a result of the impossibility to bind water [11,18]. Therefore, the structure of the fiber is more important than pH in color formation at this stage, contributing to the higher lightness of the meat.

Several authors have shown different color characteristics than those found in this research. Trocino et al. [15] found decreasing a* and b* values with the occurrence of WS, whereas no differences were found when WB appeared or directly color differences have not been significant among different myopathy levels [16]. On the contrary, Kuttapan et al. [27] found increasing b* values with both WB and WS. We have also found this tendency, increasing values as myopathy evolves towards severe levels. In the case of WS, we believe that the increased lipid accumulation [16] also incorporates pigments from the feed that are liposoluble, such as carotenoids and xanthophylls, accumulating at this levels increasing yellowness. SM has been less studied than WB or WS but has been associated with higher lipid content [10]. This would also provoke a higher accumulation of liposoluble compounds from the feed, increasing yellowness, as we have observed.

Heme proteins are mainly responsible for the red color of the muscle due to the prosthetic group with iron-containing porphyrins, being hemoglobin and myoglobin the two main heme proteins in the muscle. Reflectances at certain wavelengths have been associated with different states of myoglobin (Mb), main responsible for the red color of the muscle depending on the content and nature. Myoglobin in the muscle can be found mainly reduced as deoxymyoglobin (dMb), with an oxygen molecule as oxymyoglobin (OMb) or as metmyoglobin once the oxygen is lost (MMb). Different spectrophotometric indexes have been studied, in particular the subtraction of R630 minus R580 or the ratio R630/R580 have been associated with more redness due to either dMb or OMb [30] or even sulfmyoglobin in the case of 629 nm [31]. They are interesting indicators to assess color shelf life, reducing their value as color deteriorates. This is particularly important in red meat, but we have found that it can differentiate also different degrees of myopathies in broiler, especially R630-R580 index that had not been previously used in poultry to our knowledge. The increasing levels, as the myopathy is more severe, are indicative of increasing redness, as it has been shown in a* coordinate. Considering that the samples were measured 24 h after slaughtered and just after the breast was dissected and the skin retired, there is not time enough for color to deteriorate, which low numbers on the index would show [30]. The content of myoglobin correlates with muscle fiber composition, being the lowest in glycolytic muscles, such as *Pectoralis major*, whereas the amount of hemoglobin is quite constant in different muscles, with exceeding levels over myoglobin content in broilers [32] although these levels are not constant depending on the techniques used [32]. With a combination of spectrophotometric analysis and size-exclusion chromatography, the hemoglobin content when small hemorrhages occur, have been equivalent to hemorrhage-free muscles [32]. Petechial hemorrhagic lesions that appeared in the surface of some breasts are produced during pre-slaughter period, showing the appearance of fresh blood [27]. This hemoglobin extravasation from the veins could be easier to produce in severe myopathies due to the disorganized connective tissue and deficient circulation [27]. Therefore, although the amount of heme proteins in the muscle remain similar, the position would be different and R630-R580 index would reflect both myoglobin and hemoglobin state. This has proved sufficient to show clear differences between different myopathies degrees.

Classification of WB samples by wavelength reflectance has not been as good as of WS or SM. Reflectance seems to separate appropriately those normal samples classified as such, not being that accurate at the different degrees of WB or WS myopathy. This fact could be useful at industry level because the technique is fast and nondestructive and could help to deviate in the processing line those breasts with any degree of WS or SM towards a different commercial destination than normal breasts. Other methods have been used to discriminate normal samples without myopathies. Computer Vision System in combination with near-infrared spectroscopy has been applied to the classification of WB [33]. A 96.3% cross-validation accuracy was obtained in detecting WB samples using six wavelengths from 1261 nm onwards, although they cannot be directly compared with visible spectra. WS has been identified using radiofrequency spectra [34]. Furthermore, visible and near-infrared hyperspectral imaging has been used in discriminating WS. Jiang et al. [35] selecting seven wavelengths, among them 581 and 631, obtained a 91.7% classification rate. These studies have measured the same cranial location of the breast as in our study that has been found to be the most discriminant area [33,35]. Nevertheless, the implementation of these other methods in the poultry industry is very limited at the moment due to their cost or complexity.

### 4.3. Texture in Raw Meat

The macroscopic lesions that have been observed in the breasts (white striations parallel to the fibers, physical hardness) have also been related to microscopic lesions (loss of cross striation, regeneration of different size’s fibers, fibrosis and lipidosis) [11] affecting especially *Pectoralis major* in comparison with *Pectoralis minor*. The histological damage increases with the severity of the myopathy, and this is especially severe in wooden breast [11]. The use of a compression test where the fibers cannot expand transversally, but longitudinally in their same axis, allow us to assess the resistance of the muscle, and to individually separate the resistance due to different structures. At low compression rates (C20) the resistance is mainly due to the myofiber since the connective tissue has the property of expanding due to its elasticity, without interfering in the muscle resistance [19]. However, at higher compression rate (C80) the connective tissue is an important part in the resistance of the muscle. Severe WB has been found to provoke a 15% and 12% more resistance to the compression (C20) than normal or moderate WB breasts (Table 4), due to changes in the myofiber, whereas the presence of WS provokes an increase in the resistance of 18%. Even in whole breasts, without orientating the fibers in a perpendicular position to the applied force, a low compression rate detects differences between WB samples [36]. At a higher compression rate of C40%, Soglia et al. [37] did not find clear differences between normal and WB samples, probably because more structures other than the fiber are involved in the resistance that the muscle offers when the compression increases from 20% onwards. However, no significant changes have been found in SM myopathy, although it has to be remarked that severe SM was not found, and the number of moderate SM was fairly low with an occurrence of just 5.3% of the carcasses. Even with these low observations’ numbers, connective tissue was evidently less responsible for the resistance to compression C80 when SM appears, reflecting the disintegration of this tissue in this myopathy. Similar amount of total and soluble collagen has been found in normal or SM breasts, but with an uneven distribution of collagen thorough the muscle [10]. On the contrary, no significant differences were found for WB or WS in the independent effect of the connective tissue. At perimysium level, a thickening of the layers of connective tissue has been observed, with a diffuse thickening of the endomysial and perimysial network without increasing the amount of total collagen [16], which is the main component of connective tissue. However, other authors have found increase collagen content in the breasts with these myopathies [18]. The location of the sample within the breast also has an important impact in detecting differences, since deeper location in the WB muscle does not differ at 80% of the compression rate from normal breasts, whereas the surface WB samples are tougher [37]. The full rate of compression shows that severe WB increases the resistance of the muscle due also to the connective tissue implication. The lack of differences in WS at the full compression rate corroborates that connective tissue is not the most altered structure in this myopathy.

Texture profile analysis has been previously used to measure texture in broiler breast [18]. Higher hardness has been found in the same line as that found in the compression test for WB. In addition, severe WB and WS are less cohesive. This corresponds to the fibrosis observed as a result of the myopathy [29]. Soglia et al. [18] besides hardness, found WB and WB/WS samples to be gummier and less elastic. Our data follow the same patron, although we have not found these values significantly different between different levels of myopathy. All of these texture characteristics are the result of the different chemical changes that affect the myofibrils and the connective tissue.

### 4.4. Texture in Cooked Meat

Differences in texture for cooked meat were less evident than in raw meat. Although Warner-Bratzler is widely use in red meat, we have not found any differences and only a tendency in WS to have less resistance to the shear force as the severity of the myopathy increases. Variable amounts of loose connective tissue, together with collagen-rich connective tissue, have been found with severe myopathy [16]. Heat can solubilize these structures easier in newly formed connective tissue, making the impact of soluble connective tissue less important. A non-reducible crosslink has been described at superficial level [29]. Therefore, when heat is applied, the non-reducible collagen remains unaltered and this attains to any muscle independently of the myopathy severity. Tijare et al. [15] suggested that histological differences may impact the meaning of shear results that indirectly were used to assess tenderness related to the contractile state of the fiber.

Cooking increases hardness, adhesiveness, chewiness and reduces springiness [14]. Data in TPA have not been significantly different between levels of myopathy in cooked meat, except for WB that showed less cohesiveness, in the same way as previously found in raw meat. Although less important than in raw meat, other authors have shown differences [18] that we attribute to the different initial sampling with larger subsamples. Since differences between the superficial and deeper layers have been found in the collagen content and crosslinking [29] it can occur that smaller samples, as those use in the TPA of our work, do not include some of the structural changes that occur in the muscle, even when the sampling is intended to obtain representative equal subsamples for the different tests. Furthermore, freezing and thawing can have an impact in changing the initial structure of the muscle reducing differences, although not all quality traits seem to be affected by this process at the same rate [38].

## 5. Conclusions

Breast myopathies are an important quality problem that affect both color and texture characteristics. Each degree of severity of WB and WS of the *Pectoralis major* shows 1% difference in breast yield. Color differs between severity degrees, increasing redness and, especially, yellowness and chroma with the severity of WB, WS and SM. The subtraction R630 minus R580 has been found a useful index to differentiate breast myopathies degrees, although more research is required to assess which heme structures are implying in these differences. A modified compression test that hinders the fiber to expand transversally has been found to be an effective tool to assess hardness of different structures in WB, WS and SM myopathies. Texture differences in the myopathies are better assessed in raw than in cooked meat.

## Figures and Tables

**Table 1 foods-09-00824-t001:** Carcass quality and pH of broiler breast with different degree of myopathies.

Myopathy Degree	Wooden Breast			White Striping			Spaghetti Meat		RMSE	Significance		
	0	1	2	0	1	2	0	1		WB	WS	SM
*n*	*133*	*133*	*34*	*175*	*90*	*35*	*284*	*16*				
Carcass weight kg	1.92 b	2.01 a	2.06 a	1.92 b	2.04 a	2.06 a	1.98	1.90	0.185	<0.001	0.010	0.965
Carcass breast yield%	31.3 c	32.1 b	33.2 a	31.3 c	32.4 b	33.6 a	31.8	32.7	2.03	<0.001	<0.001	0.438
pH	5.86 b	5.90 b	6.05 a	5.86 b	5.92 ab	6.01 a	5.90	5.92	0.237	0.007	0.018	0.982

RMSE: root mean square error; a, b, c: mean values in the same row with different letters within myopathy differ significantly (*p* ≤ 0.05). WB: wooden breast; WS: white striping; SM: spaghetti meat.

**Table 2 foods-09-00824-t002:** Color of the surface of broiler breast with different degree of myopathies.

Myopathy Degree	Wooden Breast			White Striping			Spaghetti Meat		RMSE	Significance		
	0	1	2	0	1	2	0	1		WB	WS	SM
*n*	*133*	*133*	*34*	*175*	*90*	*35*	*284*	*16*				
L*	52.3 b	52.7 b	54.1 a	52.1 c	53.1 b	54.2 a	52.6	53.6	2.59	0.005	0.005	0.304
a*	−1.77 b	−1.41 a	−1.32 a	−1.74 b	−1.36 a	−1.17 a	−1.59	−1.05	0.921	0.001	0.005	0.375
b*	5.10 b	5.65 a	5.81 a	5.00 c	5.75 b	6.73 a	5.32 b	7.22 a	1.43	0.005	<0.001	0.002
Chroma*	5.57 b	6.03 a	6.11 a	5.49 c	6.08 b	6.93 a	5.75 b	7.39 a	1.32	0.042	<0.001	0.003
Hue°	110.9 a	106.1 ab	104.8 b	111.3 a	104.3 b	101.6 b	108.6	99.2	14.9	0.014	0.002	0.232
R630-R580	4.11 b	4.82 a	5.19 a	4.20 c	4.88 b	5.44 a	4.48 b	5.74 a	1.41	<0.001	<0.001	0.038
R630/R580	1.21 b	1.25 a	1.26 a	1.22	1.24	1.26	1.23	1.29	0.10	0.009	0.120	0.139

RMSE: root mean square error; a, b, c: mean values in the same row with different letters within myopathy differ significantly (*p* ≤ 0.05); R630-R580: reflectance at 630 nm minus reflectance at 580 nm; R630/R580: ratio between reflectance at 630 nm and reflectance at 580 nm.

**Table 3 foods-09-00824-t003:** Classification of breast myopathies (% of samples correctly classified) from visible spectrum (360–740 nm).

	Wavelength (nm)	Myopathy Degree			% Original Samples Correctly Classified
		0	1	2	
Wooden Breast	360, 420, 530, 580, 620, 640	64.7	53.4	2.9	52.7
White striping	360, 610, 700	92.0	25.6	14.3	63.0
Spaghetti meat	420, 630	100	0	-	94.7

**Table 4 foods-09-00824-t004:** Texture of broiler raw breast with different degree of myopathies by different texture devices.

Myopathy Degree	Wooden Breast			White Striping			Spaghetti Meat		RMSE	Significance		
	0	1	2	0	1	2	0	1		WB	WS	SM
*n*	*133*	*133*	*34*	*175*	*90*	*35*	*284*	*16*				
Compression												
C20 N/cm^2^	11.4 b	11.7 b	13.1 a	10.9 b	12.9 a	12.8 a	11.8	10.3	3.18	<0.001	0.050	0.257
C80 N/cm^2^	15.3	15.3	16.1	15.3	16.0	14.2	15.6 a	12.1 b	5.51	0.298	0.197	0.029
Ctotal N/cm^2^	22.1 b	22.8 ab	24.7 a	22.1	24.1	22.5	23.0 a	18.7 b	6.78	0.016	0.383	0.032
TPA												
Hardness N/cm^2^	11.3 b	12.1 ab	12.9 a	11.1	12.8	13.3	11.9	10.8	3.74	0.016	0.997	0.240
Adhesiveness	−47.6	−44.6	−40.7	−41.7	−50.7	−51.0	−45.5	−42.8	14.03	0.628	0.804	0.712
Cohesiveness	0.423 a	0.408 a	0.370 b	0.417 a	0.404 ab	0.385 b	0.411	0.388	0.043	0.001	0.007	0.790
Springiness cm	0.686	0.677	0.678	0.661	0.703	0.734	0.681	0.705	0.073	0.527	0.114	0.081
Gumminess N/cm^2^	4.87	5.05	4.93	4.74	5.25	5.29	4.98	4.26	1.76	0.286	0.735	0.249
Chewiness N/cm	3.42	3.48	3.35	3.18	3.74	3.97	3.45	3.01	1.27	0.258	0.844	0.599
Resilience	0.250	0.245	0.232	0.249	0.242	0.238	0.246	0.232	0.031	0.279	0.118	0.279

RMSE: root mean square error; a, b: mean values in the same row with different letters within myopathy differ significantly (*p* ≤ 0.05). TPA: texture profile analysis.

**Table 5 foods-09-00824-t005:** Texture of broiler cooked breast with different degree of myopathies by different texture devices.

Myopathy Degree	Wooden Breast			White Striping			Spaghetti Meat		RMSE	Significance		
	0	1	2	0	1	2	0	1		WB	WS	SM
*n*	*133*	*133*	*34*	*175*	*90*	*35*	*284*	*16*				
Warner–Bratzler												
WBSF kg/cm^2^	1.50	1.48	1.52	1.51	1.52	1.36	1.49	1.53	0.35	0.289	0.087	0.623
TPA												
Hardness N/cm^2^	12.9	12.9	13.1	12.7	13.3	13.2	12.9	13.3	3.01	0.985	0.403	0.478
Adhesiveness	−12.9	−12.8	−11.9	−13.0	−12.7	−11.7	−12.8	−11.9	4.92	0.906	0.309	0.626
Cohesiveness	0.515 a	0.505 a	0.475 b	0.508	0.505	0.505	0.507	0.496	0.040	0.043	0.738	0.855
Springiness cm	0.563	0.560	0.571	0.566	0.556	0.562	0.563	0.552	0.028	0.398	0.258	0.062
Gumminess N/cm^2^	6.76	6.60	6.28	6.52	6.81	6.73	6.63	6.78	1.843	0.566	0.617	0.408
Chewiness N/cm	3.81	3.70	3.60	3.70	3.80	3.80	3.74	3.76	1.069	0.674	0.749	0.606
Resilience	0.184	0.185	0.179	0.182	0.185	0.190	0.184	0.182	0.023	0.603	0.447	0.705

RMSE: root mean square error; a, b: mean values in the same row with different letters within myopathy differ significantly (*p* ≤ 0.05). WBSF: Warner-Bratzler shear force. TPA: texture profile analysis.

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
