# Peer review of "Differentiating Breast Myopathies through Color and Texture Analyses in Broiler"

_foods, 2020, doi:10.3390/foods9060824_

Round 1

Reviewer 1 Report

This paper provides information on the effects of broiler breast myopathies on colour and textural traits of the meat.

Overall, the manuscript is clear and well organized, and the experiment was well designed. However, main concern deals with lacking explanation of novelty in respect to previously published papers.

Indeed, this study includes all main broiler breast myopathies (white striping, wooden breast and spaghetti meat) according two previously published papers (Soglia et al. DOI: 10.3382/ps/pez514 and Baldi et al. 10.1016/j.foodres.2018.11.020) where colour coordinates L*a*b* and in one of the them also textural parameters (compression force) on raw and cooked meat were measured. As recently reviewed by Petracci et al. (2019), both colour and texture were assessed in many previous studies in samples affected by one or two of tested myopathies. In addition, some instrumental methods have been already proposed to discriminate normal and abnormal fillets: Traffano-Schiffo et al. (2017 - doi:10.3390/s17051024 - doi:10.1016/j.infrared.2018.11.036) proposed radiofrequency spectra, Jiang et al. 2019 - doi:10.1016/j.saa.2019.01.052) investigated hyperspectral imaging, while near-infrared spectroscopy was tested by (Geronimo et al., 2019).

Therefore, authors must clarify which is expected advancement of knowledge given by present study.

OTHER REMARKS

Lines 81-87: colour was measured on cranial portion of P. major muscles which the part more affected by myopathies. Presence of white striations in WS samples, petechiae and viscous liquid in WB samples and damaged area in SM samples, can likely affect colour readings done by colorimeter on a small area irrespective from the “true” meat colour. It must be specified how measurement were conducted in order to be maintain consistency trough the experiment and avoid any interference by the presence of each abnormality.

Lines 87-109: textural measurements on cooked samples were conducted on the right side of breasts after freezing. It is well-known that freezing can likely affect textural properties of the meat especially if meat is not frozen under industrial condition where high speed systems are used. Freezing likely can mitigate textural differences especially when normal and wooden breast is compared: this should be considered in the discussion of the results. This aspect must be taken in account also when results obtained in this study were compared with previous findings obtained using refrigerated meat.

Author Response

We want to thank the effort made by this reviewer with the comments to improve the manuscript. We hope that the responses are sufficient to clarify any doubts.

This paper provides information on the effects of broiler breast myopathies on colour and textural traits of the meat.

Overall, the manuscript is clear and well organized, and the experiment was well designed. However, main concern deals with lacking explanation of novelty in respect to previously published papers.

Indeed, this study includes all main broiler breast myopathies (white striping, wooden breast and spaghetti meat) according two previously published papers (Soglia et al. DOI: 10.3382/ps/pez514 and Baldi et al. 10.1016/j.foodres.2018.11.020) where colour coordinates L*a*b* and in one of the them also textural parameters (compression force) on raw and cooked meat were measured. As recently reviewed by Petracci et al. (2019), both colour and texture were assessed in many previous studies in samples affected by one or two of tested myopathies. In addition, some instrumental methods have been already proposed to discriminate normal and abnormal fillets: Traffano-Schiffo et al. (2017 - doi:10.3390/s17051024 - doi:10.1016/j.infrared.2018.11.036) proposed radiofrequency spectra, Jiang et al. 2019 - doi:10.1016/j.saa.2019.01.052) investigated hyperspectral imaging, while near-infrared spectroscopy was tested by (Geronimo et al., 2019).

Therefore, authors must clarify which is expected advancement of knowledge given by present study.

Breast myopathies are a hot topic in the last five years. As the reviewer has pointed out, research has been performed mainly focused on texture. But the problem is still of great magnitude for the industry and it has not been solved yet. For the first time, we have assessed texture with a modified compression device that has been used in other species, especially in ruminants, but never tried in broiler meat. Compression is a widely texture procedure, and many papers have been referred in the manuscript with results from different tests. But modifying the performance of the sample as we have done, avoiding the stretch in all directions, allows to check the response of the muscle to the compression force applied, when only the longitudinally stretch is allowed. In this way, the muscle fibre moves only in the longitudinal axis, as it would mainly move in the muscle surrounded by epimysium and perimysium. With this behaviour during the test, we can assess the force due to the fibre and to the connective tissue at different rates, and this had not been done before. This cannot happen in the tests reflected in the literature. This is a novel use of compression in broiler, and we have pointed out this aspect in L 58. Obviously, we have performed other texture tests that have been widely used by other researches, such as Warner-Bratzler or TPA, in order to compare our results.

The other focus of the manuscript is colour, and colour has been far less studied in the literature than texture to assess myopathies. We have discussed our results with the results that we have found, and the line of the discussion is not always coincidental. We have explained a theory for it, and used a new reflectance equation to differentiate myopathies. This is also novel, R630-R580 has not been previously used in broiler meat. Researches have focused on the ratio R630/R580, and we have demonstrated that the first equation is a better indicator of the degree of the myopathy. The novelty of this aspect has also been included L306.

Besides, classifying myopathies with the whole spectra has not been previously done either. We have showed those reflectances from the spectra that better fits with the classification for future studies in each myopathy.

We believe that all of this aspects are new knowledge generated by this manuscript. An introduction in the discussion has been done, as suggested by other reviewer (L209-210).

OTHER REMARKS

Lines 81-87: colour was measured on cranial portion of P. major muscles which the part more affected by myopathies. Presence of white striations in WS samples, petechiae and viscous liquid in WB samples and damaged area in SM samples, can likely affect colour readings done by colorimeter on a small area irrespective from the “true” meat colour. It must be specified how measurement were conducted in order to be maintain consistency trough the experiment and avoid any interference by the presence of each abnormality.

Colour was in triplicate the result of the average of three measurements, each in one of three different points in the surface of the Pectoralis major, avoiding dirty areas with evident haemorrhages or liquid. This has been included in the text to clarify the methodology. L 86-87.

Lines 87-109: textural measurements on cooked samples were conducted on the right side of breasts after freezing. It is well-known that freezing can likely affect textural properties of the meat especially if meat is not frozen under industrial condition where high speed systems are used. Freezing likely can mitigate textural differences especially when normal and wooden breast is compared: this should be considered in the discussion of the results. This aspect must be taken in account also when results obtained in this study were compared with previous findings obtained using refrigerated meat.

The reviewer is right in this aspect, and a remark about the possible influence of freezing has been added in L383-384, together with the reference of Soglia et al. (2019), doi:10.3382/ps/pez514. The main reason for using freezing was the load of work that had to be performed at 24 h after slaughter. Due to the time consuming technique that texture with four devices is (raw and cook meat), the same sample would not be in the same conditions if processed in fresh, since some of them would have more hours of ageing than others. So, we preferred to have all the samples under the same ageing conditions due to the high influence in texture that the enzymatic action has from the moment of the slaughter.

Reviewer 2 Report

I congratulate with the authors for the fine manuscript. English is very good, experimental design appropriate, overa presentation and discussion are very well presented.

I have only minor suggestions:

line 65, Add Spain, after Navarra

Table 2: introduce a footnote to explain R630-R580 and R630/R580

Discussion: I would start it after table 5: Furthermore, prior to discuss each paragraph, I would briefly introduce the discussion by underlying the innovative aspect of this study.

(table 5 has some trouble in the footnote)

Author Response

REVIEWER 2

We want to thank the comments made by the reviewer to improve the manuscript and are very pleased with them.

I congratulate with the authors for the fine manuscript. English is very good, experimental design appropriate, overa presentation and discussion are very well presented.

Thanks again

I have only minor suggestions:

line 65, Add Spain, after Navarra

Done (L67)

Table 2: introduce a footnote to explain R630-R580 and R630/R580

The following footnote has been included in L148: R630-R580: reflectance at 630nm minus reflectance at 580nm; R630/R580: ratio between reflectance at 630nm and reflectance at 580nm.

Discussion: I would start it after table 5: Furthermore, prior to discuss each paragraph, I would briefly introduce the discussion by underlying the innovative aspect of this study.

In this case, it has been started before Table 5 in order to save some space in the pages. But we will follow every indication of the editorial board in terms of page configuration.

The point of novelty has also been arisen by other comment from another reviewer. So, we have explicitly added that the compression device used is novel for poultry meat in L 58, and the novel use of the colour reflectance used in this study in L306. Besides, an introduction in the discussion has been added. (L209-210).

(table 5 has some trouble in the footnote)

It has been corrected, it was a problem of a space.

Reviewer 3 Report

Differentiating breast myopathies through color and 2 texture analyses in broiler

The manuscript describes 24 H postmortem evaluation of breast fillets WS, WB, and SB myopathies

Specific comments:

L23-27 Without any data or this is not an Abstract but a Summary.

L32-33 Provide a reference that documents “economic losses”.  Reference 1 only collected consumer preference on WS in 2012.

L33 Replace “provoked” with “facilitated the emergence” as in L213.

L37 Insert “in, …incorporation of nuclei into…

L38 Insert “in, …also participate in repairing…

L40 The phrase “suggesting different satellite cells number and growth kinetics than those found in low weight selection lines” is noninformational. Indicate what the difference is.

L42-43 Is this statement true for the ventral skin-surface (where scoring occurs) or just within the center of the muscle?

L46-48 Reference 10 did not report “strategies (feed presentation, nutritional composition, feeding regime)”.

L71 Replace “featherless” with “defeathered” and replace “”clawless” with “feed removed”.

L72 Provide temperature maintained during transport.

L113 Duncan’s multiple range is not the appropriated means separation scoring breast fillets into 0, 1 or 2 groups.

Table 1. The title and rows should list parameters in the same order.

L199 Table 5 indicates that cohesiveness differed for WB not SM?

L200  Table 4 indicates that raw hardness differed for WB not WS?

L205 As for L23 Provide a reference that documents “economic losses”. 

L220 Replace “increase” with “higher” or “elevated” pH, or provide data where you measured a lower pH that increase at 24h (your only pH measurement time).

L314-315 The results presented in Table 3, with 52.7% for WB (from 2.9 to 64.7%) and 63% for WS (from 14.3 to 92%) as “correctly classified” explain how this level of low level precision in classification could be usable. The R630-R580 indexes appear to be able to separate 0-SB from 1-SB in the absence of 2-SB muscle.

L372 Provide a correlation r^2 value to support the 1% increase in breast yield.

Author Response

REVIEWER 3

We want to thank the comments made by the reviewer to improve the manuscript. We hope that the responses are sufficient to clarify any doubts.

Differentiating breast myopathies through color and 2 texture analyses in broiler

The manuscript describes 24 H postmortem evaluation of breast fillets WS, WB, and SB myopathies

Specific comments:

L23-27 Without any data or this is not an Abstract but a Summary.

Some color data have been added to the abstract (L24-24) but the extension is very restricted in this Journal and cannot fit more words without substantially changing the structure.

L32-33 Provide a reference that documents “economic losses”.  Reference 1 only collected consumer preference on WS in 2012.

Kuttapan et al. (2016) has been included in this statement.

L33 Replace “provoked” with “facilitated the emergence” as in L213.

Done

L37 Insert “in, …incorporation of nuclei into…

Done

L38 Insert “in, …also participate in repairing…

Done

L40 The phrase “suggesting different satellite cells number and growth kinetics than those found in low weight selection lines” is noninformational. Indicate what the difference is.

“..higher satellite cells number and faster growth kinetics..” has been added to explain the difference. L41.

L42-43 Is this statement true for the ventral skin-surface (where scoring occurs) or just within the center of the muscle?

This statement is based on Geiger et al. (2018), who assessed two lines of broilers selected for high weight and low weight. In their experiment, they removed the outer layer of the Pectoralis major muscle from the pieces that were initially collected and frozen. So I cannot assure that the statement is true for the surface of the muscle. An indication towards this aspect has been added (L42).

L46-48 Reference 10 did not report “strategies (feed presentation, nutritional composition, feeding regime)”.

It has been changed to Kuttapan et al. (2012).

L71 Replace “featherless” with “defeathered” and replace “”clawless” with “feed removed”.

Done, the sentence has been amended to ‘Carcasses were defeathered, eviscerated, headless and vent after feed removal prior to chilling’ (L73).

 L72 Provide temperature maintained during transport.

4ºC has been included (L74).

L113 Duncan’s multiple range is not the appropriated means separation scoring breast fillets into 0, 1 or 2 groups.

We have reanalysed the data and Tukey multiple range test has been used.

Table 1. The title and rows should list parameters in the same order.

Carcass quality and pH have been reorder (L127).

L199 Table 5 indicates that cohesiveness differed for WB not SM?

It has been corrected, it was a mistake in the text and SM has been changed to WB (L204).

L200  Table 4 indicates that raw hardness differed for WB not WS?

In this case, L200 refers to aspects in cooked meat, whereas Table 4 is based on raw meat. I think there is a confusion in this comment.

L205 As for L23 Provide a reference that documents “economic losses”. 

Kuttapan et al. (2016) has been added as a reference (L212).

L220 Replace “increase” with “higher” or “elevated” pH, or provide data where you measured a lower pH that increase at 24h (your only pH measurement time).

Measuring the evolution of pH through ageing was not an aim of this work. ‘Higher’ has been added, clarifying the sentence since severe WS shows higher pH than normal breasts (L227).

L314-315 The results presented in Table 3, with 52.7% for WB (from 2.9 to 64.7%) and 63% for WS (from 14.3 to 92%) as “correctly classified” explain how this level of low level precision in classification could be usable. The R630-R580 indexes appear to be able to separate 0-SB from 1-SB in the absence of 2-SB muscle.

The following explanation has been added: ‘Classification of WB samples by wavelength reflectance has not been as good as of WS or SM. Reflectance seems to separate appropriately those normal samples classified as such, not being that accurate at the different degrees of WB or WS myopathy. This fact could be useful at industry level because the technique is fast and non-destructive, and could help to deviate in the processing line those breasts with any degree of WS or SM towards a different commercial destination than normal breasts’ (L323-328).

L372 Provide a correlation r^2 value to support the 1% increase in breast yield.

Correlation has been provided in L157 (r=0.28 and r=0.35, respectively, P<0.001). The wording in the conclusion has changed to: Each degree of severity of WB and WS of the Pectoralis major shows 1% difference in breast yield (L387-388).

Round 2

Reviewer 1 Report

Revision is not satisfactory for the following reasons:

  1. Some previous papers used compression tests to assess textural differences in broiler breast meat affected by myopathies:

Sun et al. (2018 - https://doi.org/10.3382/ps/pey107) - Instrumental compression force and meat attribute changes in woody broiler breast fillets during short-term storage

Soglia et al. (2017- https://doi.org/10.3382/ps/pex115) – Superficial and deep changes of histology, texture and particle size distribution in broiler wooden breast muscle during refrigerated storage

Aguirre et al. (2018 - https://doi.org/10.3382/ps/pex428) - Descriptive sensory and instrumental texture profile analysis of woody breast in marinated chicken

However, authors stated that “compression test…… will be used for the first time in poultry” (lines 57-58). In example, Soglia et al. (2017) used compression rate of 40% and 80%, very similar to those adopted in the present study (20 and 80%). It is not clear the novelty of the analytical approach adopted in this study.

  1. As reported in the previous revision round, some instrumental methods have been already proposed to discriminate normal and abnormal fillets: Traffano-Schiffo et al. (2017 - doi:10.3390/s17051024 - doi:10.1016/j.infrared.2018.11.036) proposed radiofrequency spectra, Jiang et al. 2019 - doi:10.1016/j.saa.2019.01.052) investigated hyperspectral imaging, while near-infrared spectroscopy was tested by (Geronimo et al., 2019). Comparison with previously proposed methods should be discussed in respect to the colorimeter method proposed in the present study. Otherwise, relevance of results found in this study seems weak.

Author Response

We believe that the discussion has improved with the new amendments, and expect this revision will be now satisfactory.

Revision is not satisfactory for the following reasons:

  1. Some previous papers used compression tests to assess textural differences in broiler breast meat affected by myopathies:

Sun et al. (2018 - https://doi.org/10.3382/ps/pey107) - Instrumental compression force and meat attribute changes in woody broiler breast fillets during short-term storage

Soglia et al. (2017- https://doi.org/10.3382/ps/pex115) – Superficial and deep changes of histology, texture and particle size distribution in broiler wooden breast muscle during refrigerated storage

Aguirre et al. (2018 - https://doi.org/10.3382/ps/pex428) - Descriptive sensory and instrumental texture profile analysis of woody breast in marinated chicken

However, authors stated that “compression test…… will be used for the first time in poultry” (lines 57-58). In example, Soglia et al. (2017) used compression rate of 40% and 80%, very similar to those adopted in the present study (20 and 80%). It is not clear the novelty of the analytical approach adopted in this study.

The reviewer is right about the previous use of compression tests to analyse texture in broiler, with different cells and devices. We have revised the manuscript, and mention to the first time, that was added after the first revision, has been deleted. I must admit that I follow the work in poultry by M. Petracci’s group, and this paper by Soglia et al. had escaped from our manuscript, even than other papers by the same author and the same research group were being referred. One of the devices that is used in that manuscript is precisely the same device that we have used, and the manuscript even refers to us in the methodology as the original source. Nevertheless, Soglia et al. (2017) do not consider the compression at a rate of 20%, but at 40%, that had already been proved in the original paper (Campo et al., 2000) less discriminant for the muscle fiber than 20%, which is the one we have studied. In fact, at 24h post-mortem, which is the ageing of our samples, Soglia et al (2017) did not find a clear difference between normal and WB samples, depending on the location of the analysis. This has been added to the discussion (L354-357, 368-370) and makes a clear difference between the two studies. Besides, Soglia et al (2017) only focused on WB myopathy, whereas we have applied this device also to WS and SM myopathies.

Sun et al (2018), instead, have used a compression at 20%, which is the same ratio that we have used, but with a different methodology and a non-modified device. They have not sampled the breast in sub samples with parallel fibers perpendicular to the direction of the applied force. Besides, there is not a wall that hinders the transversal expansion. They have used the whole breast with the aim of comparing this analysis to a tactile evaluation. A very different approach than what we have performed, which is assessing the implication of different structures of the muscle. A mention to this fact has been added in the discussion (L352-354). On top of this, they have not assessed WS or SM, which we have. In both cases (Soglia et al., 2017; Sun et al., 2018), WS and SM were not analysed, a clear difference with our study.

However, we disagree with the work of Aguirre et al. (2018) being comparable to ours. They have used marinated chicken, not fresh breast. Marinate modifies the structure of the muscle and the use of sensory analysis is not part of the subject of the issue were this paper is aiming to be published. Other manuscripts with the same compression tests but in fresh meat had been already discussed in the manuscript. Therefore, we do not consider adequate to include texture of marinated meat as part of the discussion.

  1. As reported in the previous revision round, some instrumental methods have been already proposed to discriminate normal and abnormal fillets: Traffano-Schiffo et al. (2017 - doi:10.3390/s17051024 - doi:10.1016/j.infrared.2018.11.036) proposed radiofrequency spectra, Jiang et al. 2019 - doi:10.1016/j.saa.2019.01.052) investigated hyperspectral imaging, while near-infrared spectroscopy was tested by (Geronimo et al., 2019). Comparison with previously proposed methods should be discussed in respect to the colorimeter method proposed in the present study. Otherwise, relevance of results found in this study seems weak.

Comparison with some of these methods has been made (L329-338). We did not choose to discuss other methods that were not used in the study in our first version, trying to focus on reflectances obtained with a spectrocolorimeter, and a mention to this aim has been added in L60-61. This equipment is available in most labs that study meat quality al colour level. Therefore, our results would be easily compared and contrasted by other researches. It is also available in the industry, and the results of this manuscript are already being implemented in the industry at abattoir level, because the machines are easy to use, fast in use and costly affordable. The references that have been included in the discussion did not assess indexes to differentiate myopathies, and only focused either on WB or WS assessments, whereas we have studied three myopathies on the same manuscript. We have found the subtract R630 minus R580 a very useful index for discriminating myopathy degrees, and this is novel and should be considered for publication.

There are methodologies other than spectrocolorimetry that can be used with the same aim as in our manuscript. VIS-NIR is also a widely used technique, implemented in the industry especially when measuring materials with low moisture, where its effectiveness is very high. However, hyperspectral imaging and radiofrequency are less used, and, to our knowledge, they are not implemented at industry level in our country for the aim of the paper. Neither is computer vision system associated to one of the former methods. There are only two abattoirs in Spain that uses CVS for classifying beef, and none in poultry due to the complexity and cost of the system. The same attains for a hyperspectral camera, especially when the range is VIS-NIR. I think there is a wide area of research and lots of work have to be performed before the competitiveness of the systems makes them useful for the industry. We understand the point of view of the reviewer, but the aim of this manuscript was never to develop new methods, but to use two existing ones with a different approach, always with the objective of being used at practical level. There is room for another manuscript just by discussing all methods for assessment of myopathies.
